# The Role of Fatty Acid Synthase in the Vascular Smooth Muscle Cell to Foam Cell Transition

**DOI:** 10.3390/cells13080658

**Published:** 2024-04-09

**Authors:** Bethany J. Bogan, Holly C. Williams, Claire M. Holden, Vraj Patel, Giji Joseph, Christopher Fierro, Hugo Sepulveda, W. Robert Taylor, Amir Rezvan, Alejandra San Martin

**Affiliations:** 1Department of Medicine, Division of Cardiology, Emory University, Atlanta, GA 30322, USA; bethany.joyce.bogan@emory.edu (B.J.B.); hcwilli@emory.edu (H.C.W.); claire.holden@emory.edu (C.M.H.); vrajtpatel7@gmail.com (V.P.); gjosep2@emory.edu (G.J.); w.robert.taylor@emory.edu (W.R.T.); arezva2@emory.edu (A.R.); 2Institute of Biomedical Sciences, Faculty of Medicine, Universidad Andres Bello, Santiago 8370071, Chile; cristopher.fierro@unab.cl (C.F.); hugo.sepulveda@unab.cl (H.S.)

**Keywords:** VSMC phenotype, fatty acid synthase (FASN), Krüppel-like factor 4 (KLF4), foam cells, lipids, lipid droplets, esterification

## Abstract

Vascular smooth muscle cells (VSMCs), in their contractile and differentiated state, are fundamental for maintaining vascular function. Upon exposure to cholesterol (CHO), VSMCs undergo dedifferentiation, adopting characteristics of foam cells—lipid-laden, macrophage-like cells pivotal in atherosclerotic plaque formation. CHO uptake by VSMCs leads to two primary pathways: ABCA1-mediated efflux or storage in lipid droplets as cholesterol esters (CEs). CE formation, involving the condensation of free CHO and fatty acids, is catalyzed by sterol O-acyltransferase 1 (SOAT1). The necessary fatty acids are synthesized by the lipogenic enzyme fatty acid synthase (FASN), which we found to be upregulated in atherosclerotic human coronary arteries. This observation led us to hypothesize that FASN-mediated fatty acid biosynthesis is crucial in the transformation of VSMCs into foam cells. Our study reveals that CHO treatment upregulates FASN in human aortic SMCs, concurrent with increased expression of CD68 and upregulation of KLF4, markers associated with the foam cell transition. Crucially, downregulation of FASN inhibits the CHO-induced upregulation of CD68 and KLF4 in VSMCs. Additionally, FASN-deficient VSMCs exhibit hindered lipid accumulation and an impaired transition to the foam cell phenotype following CHO exposure, while the addition of the fatty acid palmitate, the main FASN product, exacerbates this transition. FASN-deficient cells also show decreased SOAT1 expression and elevated ABCA1. Notably, similar effects are observed in KLF4-deficient cells. Our findings demonstrate that FASN plays an essential role in the CHO-induced upregulation of KLF4 and the VSMC to foam cell transition and suggest that targeting FASN could be a novel therapeutic strategy to regulate VSMC phenotypic modulation.

## 1. Introduction

Vascular smooth muscle cells (VSMCs) are integral to the structural integrity and function of blood vessel walls. Differentiated VSMCs express genes related to the contractile apparatus, such as MYH11, TAGLN, CNN1, and ACTA2 [1]. These cells exhibit a high propensity for phenotypic modulation, even in adults, in response to stimuli like cholesterol or growth factors [2], shifting from a contractile, differentiated phenotype to a synthetic/proliferative one or acquiring characteristics of other cell lineages [2].

Atherosclerosis is a prevalent chronic inflammatory disease initiated by damage to the endothelial cell layer [3,4] and is characterized by the accumulation of fatty intimal deposits, or atheromatous plaques, within large and intermediate arteries [5]. A main component of the atheromatous plaques are foam cells, a type of macrophage capable of amassing large intracellular lipid pools [6]. Thus, foam cells and their precursors are crucial in the development of atherosclerosis. 

While the cellular origins of foam cells are not fully elucidated, lineage tracing studies in mice suggest that they arise from diverse cell lineages such as monocytes/macrophages, endothelial cells, and VSMCs [4], with VSMC-derived foam cells forming a significant component of human atherosclerotic lesions [7,8,9].

Importantly, in atherosclerosis, a substantial number of VSMCs undergo phenotypic changes, downregulating contractile genes while upregulating genes that promote cellular plasticity and foam cell conversion [7,8,9]. This phenotypic shift includes the uptake and storage of lipids and an increase in the expression of Kruppel-like factor 4 (KLF4), a transcriptional regulator of VSMC plasticity [10,11], and macrophage markers such as CD68 [12]. Although it has been established that VSMCs contribute significantly to foam cell formation, the precise mechanisms by which VSMC-derived foam cells acquire and store intracellular lipids are not yet fully understood.

The regulation of intracellular cholesterol concentrations hinges on four key processes: efflux, uptake, storage, and biosynthesis [13]. Upon internalization, free cholesterol can be either exported by the ATP-binding cassette transporter ABCA1 [14,15] or esterified by the enzyme sterol O-acyltransferase 1 (SOAT1). This esterification mitigates the cytotoxicity of free cholesterol and enables its storage in lipid droplets [16]. In this process, SOAT1 transfers a fatty acid to the hydroxyl group of cholesterol, forming a cholesterol ester. Notably, an increase in vascular fatty acid synthesis, particularly in atherosclerotic plaques, is associated with the disease [17]. Evidence suggests that reducing fatty acid synthase (FASN) in macrophages markedly decreases plaque formation in atherosclerosis models [17]. However, the role of lipogenesis in VSMC-derived foam cell formation has not been investigated. In the current study, we demonstrate the necessity of FASN expression in this process, involving a mechanism that regulates cholesterol efflux transporters. This research potentially uncovers new aspects of VSMC functionality in atherosclerosis, contributing to a deeper understanding of the disease’s molecular underpinnings.

## 2. Materials and Methods

### 2.1. Human Coronary Arteries 

Human coronary arteries were collected from patients with end-stage heart failure undergoing heart transplantation at Emory University Hospital, as described previously [18]. All procedures were approved by the Emory Institutional Review Board. Briefly, hearts were collected during surgery, placed in ice-cold normal saline, and transferred immediately to an ice-cold Krebs buffer. The left anterior descending, left circumflex, and right coronaries were excised and fixed in 10% buffered formaldehyde for up to three days, followed by 70% ethanol. Tissues were embedded in paraffin before sectioning and processing for morphology and FASN staining. Images were taken using a Zeiss LSM 510 META confocal microscope with ×20, ×40, or ×63 lenses. The severity of atherosclerosis was ranked based on histological classification as described previously [19], and sections of stages II and IV were used in this study. Stage II arteries exhibited established neointima with high smooth muscle cell density but without a lipid core; stage IV arteries exhibited an established lipid core but no thrombosis [19].

### 2.2. RNA-Seq in Cholesterol-Treated HASMCs

Bulk RNA-Seq data from human aortic smooth muscle cells treated with cholesterol were downloaded from Gene Expression Omnibus (GSE181362) [20] and re-analyzed. To perform this experiment, cholesterol was loaded into cultured human aortic smooth muscle cells (HASMCs) (ScienCell Research Laboratories, cat # 6110) through water-soluble Cholesterol-methyl-β-cyclodextrin (Chol:MβCD, Sigma Aldrich, St. Louis, MO, USA), C4951) to a final concentration of 10 μg/mL. Total mRNA from four independent samples was purified using oligo(dT)-attached magnetic beads, fragmented, and then converted into double-stranded cDNA through reverse transcription and second-strand synthesis. After end repair with A-Tailing Mix and RNA Index Adapters, the cDNA was PCR amplified, purified, and checked for quality on an Agilent Technologies 2100 bioanalyzer. The PCR products were then circularized to form a library, which was amplified to create DNA nanoballs. These nanoballs were loaded onto a nanoarray for single-end 50-base read generation on the BGIseq500 platform.

For the alignment of RNA sequencing reads to the reference genome, the STAR software (version 2.7.6a) was employed. Subsequent counting and analysis of RNA sequencing reads were performed using featureCounts (version 1.6.3). The analysis for identifying differentially expressed genes and adjusted *p* values was performed using Deseq2 [21]. Criteria for considering genes as differentially expressed included a fold change greater than 2 and *p*-adjusted values below 0.05.

### 2.3. Cell Culture and Cholesterol Treatment

Human aortic smooth muscle cells (HASMCs) were isolated from human aortic tissue and cryopreserved at passage three by a commercial source (Thermo Fisher Scientific, Waltham, MA, USA cat # C0075C). HASMCs were grown as recommended by the vendor in basal media 231, with the addition of growth supplements. After cells reached ~80% confluence, growth supplements were removed 24 h prior to experiments. Cells were incubated with water-soluble Cholesterol-methyl-β-cyclodextrin (Chol:MβCD, MilliporeSigma, St. Louis, MO, USA, catalog# C4951) to a final concentration of 10 μg/mL.

### 2.4. Cell Transfection

Cells were transfected using Lipofectamine RNAiMAX reagent (Thermo Fisher Scientific, Waltham, MA, USA) following the manufacturer’s suggested protocol. Sequences were FASN sense 5′-CCGCCATGCAACGGGATTGAA-3′ and KLF4 sense 5′-TTGGTGAGTCTTGGTTCTAAA-3′. AllStars Negative Control siRNA (Qiagen, Germantown, MD, USA, cat#1027281) was used as a control siRNA.

### 2.5. Preparation of Bovine Serum Albumin (BSA)–Palmitate Conjugate and Cell Treatment

The fatty acid-free (FAF) BSA–palmitate solution was prepared following the methods previously described [22,23] with slight modifications. Initially, BSA was dissolved in a 150 mM NaCl solution, then filtered and further diluted with the same NaCl solution. Sodium palmitate (P9767 Sigma-Aldrich) was also dissolved in a 150 mM NaCl solution. Upon complete dissolution, the sodium palmitate solution was gradually added to the BSA solution in 5 mL increments. Following this addition, the pH of the solution was adjusted to 7.4. This process of conjugating FAF BSA with palmitate aimed to enhance the palmitate’s solubility. For control purposes, a corresponding BSA solution mixed with NaCl was utilized.

HASMCs were cultured according to the recommendations provided by the supplier, using basal media 231, supplemented with specific growth additives. Once the cells achieved approximately 80% confluence, the growth supplements were withdrawn 24 h before initiating the experiments. The cells were then plated in 6-well plates and incubated for 72 h with sodium palmitate at a final concentration of 1 μM, with or without the addition of cholesterol.

### 2.6. Intracellular Neutral Lipid Staining

Intracellular neutral lipids were visualized using Oil Red O staining after exposing cells to Chol:MβCD. Following treatment, cells underwent a series of washes, initially with phosphate-buffered saline (PBS) and subsequently fixed with 10% formalin. Post-fixation, cells were washed with deionized water and then with 60% isopropanol to prepare for staining.

The Oil Red O stock solution was prepared by dissolving 0.35 g of Oil Red O powder in 100 mL of isopropanol, followed by filtration through a 0.22 μm syringe filter to ensure clarity. This stock solution was then diluted to create the working solution using a mixture of three parts Oil Red O stock and two parts distilled water. The resulting working solution was filtered again prior to application to the cells.

Staining was performed by applying the filtered working solution directly to the cells, which were then imaged using an Olympus microscope. To quantify the staining, images were analyzed with ImageJ software, version 1.54h. The process involved selecting cells within the field of view and inverting the color to highlight intracellular red staining distinctly. The background was cleared to eliminate any non-specific staining from the quantitative analysis.

The quantification was based on the percentage area of red staining within each image. This was achieved by setting a numerical threshold specific to the red staining intensity in the cells. The ratio of detected red particles to the total cell area was calculated, maintaining a consistent threshold across all images to ensure uniformity in measurement. ImageJ software then computed the percentage area of red staining in the cells based on the predefined threshold. All images were captured at 20× magnification, facilitating detailed observation and analysis of the stained intracellular lipids.

### 2.7. Western Blots

Total protein lysates were prepared in RIPA buffer plus a protease/phosphatase inhibitor cocktail. Protein samples were separated on Express-Plus Page Gels (GenScript, Piscataway, NJ, USA) with Tris-MOPS buffer containing SDS and transferred onto an Immobilon-p membrane (MilliporeSigma, Burlington, MA, USA, catalog# IPVH00010). All antibodies were incubated for 24 h at 4 °C. Protein bands were visualized using HRP-conjugated secondary antibodies and an ECL western blotting substrate (Luminata Crescendo or Forte, MilliporeSigma, Burlington, MA, USA) and imaged on a Kodak camera system and a Bio-Rad ChemiDoc system. The antibodies used for this study can be found in Appendix A.

### 2.8. Statistical Analysis 

Data are presented as mean ± SEM from independent experiments. Significance was determined using a Student’s *t*-test for unpaired samples, a one-way ANOVA, or a two-way ANOVA, followed by Bonferroni’s post hoc test for multiple comparisons. GraphPad Prism 9 was used for statistical analysis. A threshold of *p* < 0.05 was considered significant and is indicated by * *p* < 0.05, ** *p* < 0.01, *** *p* < 0.001, and **** *p* < 0.0001.

## 3. Results

### 3.1. Cholesterol-Induced Phenotypic Switching Is Accompanied by the Upregulation of the Fatty Acid Synthase (FASN) in VSMC In Vivo and In Vitro

To initiate an exploration into the possible involvement of fatty acid synthase (FASN) in the phenotypic transition of vascular smooth muscle cells (VSMCs) into foam cells, we commenced by examining FASN expression in human atherosclerotic arteries. Utilizing a biobank, these arteries were classified into various stages of atherosclerosis based on hematoxylin and eosin (H&E) staining [19]. For our investigation, we selected arteries corresponding to stages II and IV of the disease. Notably, in stage II arteries, FASN expression was almost undetectable, whereas in more advanced stage IV, there was a marked increase in FASN expression observed both in the arterial media and the neointima, as illustrated in Figure 1.

When VSMCs are loaded with cholesterol, they begin to form lipid droplets and transition into foam cells as they uptake an abundance of lipids [24]. Since we hypothesized that lipogenesis may play a pivotal role in VSMC-derived foam cell formation, we used available databases of RNAseq of the HASMCs treated with cholesterol to investigate if the cells upregulated transcripts from enzymes belonging to the de novo lipogenesis (DNL) pathway in VSMCs as they transitioned to a foam cell phenotype. We found that, compared to mRNA transcripts from the control group, cholesterol-treated HASMCs significantly upregulate FASN, while other members of the DNL pathway, such as ACACA and ACLY, were unchanged (Figure 2A,B). This discovery warranted further investigation. To confirm these results, we treated HASMCs with 10 µg/mL Chol:MβCD and evaluated protein expression under the same conditions. We examined FASN protein expression along with foam cell markers and smooth muscle differentiation genes, allowing us to assess the transition of VSMC to the VSMC-derived foam cell phenotype induced by the cholesterol treatment. Consistent with the RNAseq data (Figure 2), cholesterol treatment significantly increased FASN expression (Figure 3). Furthermore, the increase in FASN protein expression overlapped with the marked decrease in expression levels of VSMC contractile genes (ACTA2, CNN, and TAGLN) and the concomitant upregulation of foam cell markers CD68 and KLF4 (Figure 3).

The findings indicate that an upregulation of FASN is associated with the cholesterol-induced phenotypic transition of VSMCs. This upregulation may be a critical factor in the VSMC’s response to cholesterol and their subsequent phenotypic transformation. Consequently, we conducted further research to examine the effects of FASN depletion on the formation of foam cells derived from VSMCs.

### 3.2. FASN Deficiency Inhibits Cholesterol-Induced VSMC-Derived Foam Cell Formation

In our subsequent investigation, we focused on the effects of FASN deficiency on VSMC phenotypic alteration following cholesterol exposure. For this purpose, HASMCs were transfected with FASN-targeting siRNA (siFASN) or a control (siNegative), followed by a 72 h cholesterol treatment. Remarkably, western blot analysis revealed that the absence of FASN significantly inhibited the cholesterol-induced foam cell markers CD68 (Figure 4A) and the proatherogenic transcription factor KLF4 (Figure 4B) when compared to the control siRNA group treated with cholesterol. This notable reduction in CD68 and KLF4 expression in FASN-deficient cells prompted us to further explore how FASN deficiency influences the buildup of intracellular lipids in VSMC-derived foam cells.

To this end, cells were transfected with either siFASN or siNegative and exposed to Chol:MβCD. Subsequently, the accumulation of lipid droplets was assessed using Oil Red O staining, a method used for staining and visualizing neutral lipids. Baseline staining in both siFASN and siNegative groups was minimal. However, there was a pronounced increase in Oil Red O staining in the siNegative group following cholesterol treatment, whereas the siFASN group exhibited staining levels similar to baseline, as depicted in Figure 5. The diminished lipid accumulation in FASN-deficient cells highlights the necessity of FASN expression for the internal buildup of lipids during the formation of foam cells in VSMCs.

These observations indicate a critical interplay between FASN-mediated lipogenesis and the transformation of smooth muscle cells into foam cells. Supporting this concept, we noted that in the presence of palmitate (a principal catalytic product of FASN), the accumulation of intracellular lipids and the establishment of the foam cell phenotype are markedly exacerbated (Figure 6). This finding further underscores the essential role of de novo lipogenesis in the development of the foam cell phenotype.

### 3.3. FASN Regulates Proteins from the Cholesterol Efflux Pathway

As previously discussed, cellular cholesterol storage is predominantly governed by mechanisms involving cholesterol esterification and efflux. Specifically, free cholesterol, which is not subject to esterification by enzymes like SOAT1, is effluxed from the cell via the ATP-binding cassette transporter A1 (ABCA1) cholesterol transporter. In VSMCs, exposure to cholesterol notably upregulates both ABCA1 and SOAT1 in untreated cells (as shown in Figure 7A) and in cells transfected with siNegative, as depicted in Figure 6B,C. Intriguingly, in FASN-deficient VSMCs, cholesterol treatment not only significantly enhances ABCA1 expression (Figure 7B) but also impedes the cholesterol-induced surge in SOAT1 expression (Figure 7C). This augmented expression of the ABCA1 transporter, coupled with the suppression of SOAT1 transferase activity, suggests that a lack of FASN activates signaling pathways that favor the efflux of free cholesterol as opposed to esterification and storage. Such modifications in cholesterol efflux enzymes in FASN-deficient VSMCs are consistent with the observed reductions in lipid accumulation within these cells, indicating a vital interplay between FASN activity, lipid handling, and cholesterol homeostasis in VSMCs. 

### 3.4. KLF4 Is the Downstream Effector of FASN

After establishing the role of FASN in the cholesterol-induced transformation of VSMCs into foam cells and observing the alterations in cholesterol esterification and efflux markers resulting from FASN knockdown via siRNA, we began to explore the potential linkage between FASN-mediated metabolic changes and the expression of the transcription KLF4, which is activated in a FASN-dependent manner (Figure 4B). Given KLF4’s extensive role in the shift of VSMCs from their contractile phenotype, we hypothesized a more significant connection between FASN and KLF4.

In pursuit of this hypothesis, we conducted experiments involving siRNA-mediated knockdown of KLF4 in conjunction with cholesterol treatment. We observed that this knockdown markedly suppressed the expression of SOAT1, a pattern mirroring that seen in HASMCs with combined cholesterol treatment and FASN deficiency (Figure 8). Similarly, an investigation into the expression of the ABCA1 transporter under siKLF4 and cholesterol conditions revealed a significant increase in ABCA1 expression (Figure 8).

These findings suggest a mechanistic pathway where FASN regulates KLF4 expression. Both elements appear to be crucial in inhibiting SOAT1, a vital cholesterol esterification enzyme within smooth muscle cells. This inhibition leads to reduced lipid droplet formation, thereby influencing the transformation of smooth muscle cells into foam cells. In essence, our results indicate that FASN plays a pivotal role in regulating the VSMC phenotypic shift towards a foam cell phenotype in a KL4-dependent manner (Figure 9).

## 4. Discussion

The phenotypic switch that results from foam cell formation is a paramount contributor to atherosclerotic plaque. Maintaining a homeostatic relationship between cholesterol esterification and cholesterol efflux allows for a healthy balance between intracellular cholesterol esters and free cholesterol leaving the cell through ABCA1 [25]. Our results demonstrate that manipulating the key proteins that regulate cholesterol esterification and cholesterol efflux through the downregulation of FASN and KLF4 displays a trend that could increase cholesterol efflux and result in a slower progression of atherosclerosis. 

Our findings indicate that FASN expression increases with cholesterol treatment and as atherosclerotic disease advances in humans. Furthermore, we demonstrate that FASN is essential for the transition of smooth muscle cells into foam cells, offering a potential therapeutic target for atherosclerosis. Consistent with FASN’s role in regulating VSMC phenotype, a recent study by Cao et al. found that FASN expression significantly promotes PDGF-induced VSMC proliferation and neointima formation [26].

In our study, a lack of FASN expression inhibited the development of cholesterol-induced foam cells derived from smooth muscle cells. The macrophage and macrophage-like markers CD68 and KLF4 were used as determinants of vascular smooth muscle cells undergoing a phenotypic switch similar to that of a differentiated monocyte. The differentiated monocytes are found as macrophages in the atherosclerotic plaque and acquire lipids via phagocytosis; smooth muscle cell-derived foam cells have been found to acquire similar characteristics. Smooth muscle cells treated with cholesterol express the same macrophage markers and undergo a phenotypic switch to foam cells. When FASN expression was reduced in vascular smooth muscle cells, it was found that the expression of CD68 and KLF4 was decreased, even with the treatment of cholesterol. Another foam cell marker affected by a change in FASN expression was SOAT1, a mediator of cholesterol esterification. This reduction in expression connects the smooth muscle cell to the foam cell transition and fatty acid biosynthesis. We believe that FASN could be a key component of the VSMC to foam cell phenotypic switching mechanism. This idea is further conveyed by our Oil Red O results, which display a decrease in the accumulation of lipids in FASN-deficient VSMCs treated with cholesterol. Because Oil Red O stains neutral intracellular lipids, staining is an indicator of esterified (stored) cholesterol. The visible difference in lipid accumulation between FASN-deficient and control VSMCs is significant (Figure 5). These data support our claim that FASN is a major regulatory factor in the accumulation and esterification of neutral lipids, such as cholesterol esters, in smooth muscle cell-derived foam cells.

Furthermore, a reduction in FASN increased the expression of ABCA1, which is beneficial for cellular lipid accumulation in terms of less cholesterol esterification and more cholesterol efflux as free cholesterol. These results show a decrease in cholesterol ester storage as intracellular lipid droplets and an increase in the expression of ABCA1, a major efflux transporter, due to a disruption in the DNL pathway. This suggests FASN-mediated cholesterol esterification as a major mechanism that prevents VSMCs from becoming overloaded with lipids, making them less likely to transition into foam cells. Similar results were observed in KLF4-deficient VSMCs, where cholesterol treatment showed a decrease in the foam cell marker SOAT1 and an increase in the marker of cholesterol efflux ABCA1. This indicates that the metabolic changes witnessed by the deficiency of FASN could be mediated by VSMC-specific changes in KLF4 expression. FASN is the key regulating factor of a mechanism that results in an anti-foam cell pathway at the protein level.

The cellular modifications seen as a result of a FASN and KLF4 knockdown have consistently displayed a novel mechanism connecting fatty acid biosynthesis and the formation of smooth muscle cell-derived foam cells after undergoing a cholesterol-induced phenotypic switch. Our data suggest that FASN and KLF4 participate in the esterification of cholesterol and foam cell formation and may contribute to atherosclerotic plaque accumulation. This reveals FASN deficiency as a key nexus of therapeutic potential.

These results support the idea that the transition of smooth muscle cells to foam cells requires the upregulation of FASN, which could be a potential drug target to treat atherosclerosis. The therapeutic potential of FASN is centered on the connection between fatty acid biosynthesis and the inverse relationship between cholesterol esterification and efflux.

## Figures and Tables

**Figure 1 cells-13-00658-f001:**
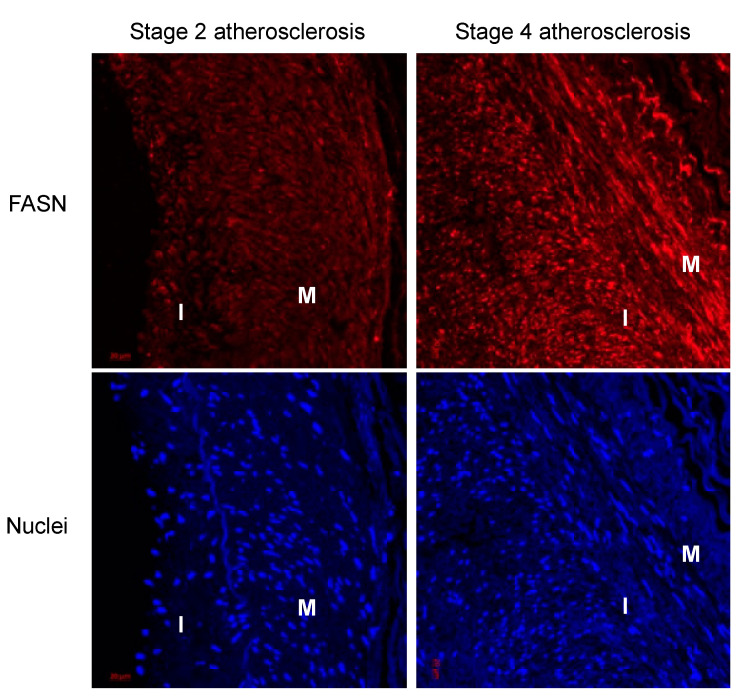
FASN expression is increased in atherosclerotic human coronary arteries as plaque formation progresses. FASN (red) expression in the media and intimal lesions is higher in stage 4 than stage 2 atherosclerotic lesions. Blue: DAPI staining of nuclei. Media (M) and Intima (I) are shown. Tissue samples were collected from the human biobank of atherosclerosis samples available in Emory University’s cardiology department. Images representative of 3–4 independent patients.

**Figure 2 cells-13-00658-f002:**
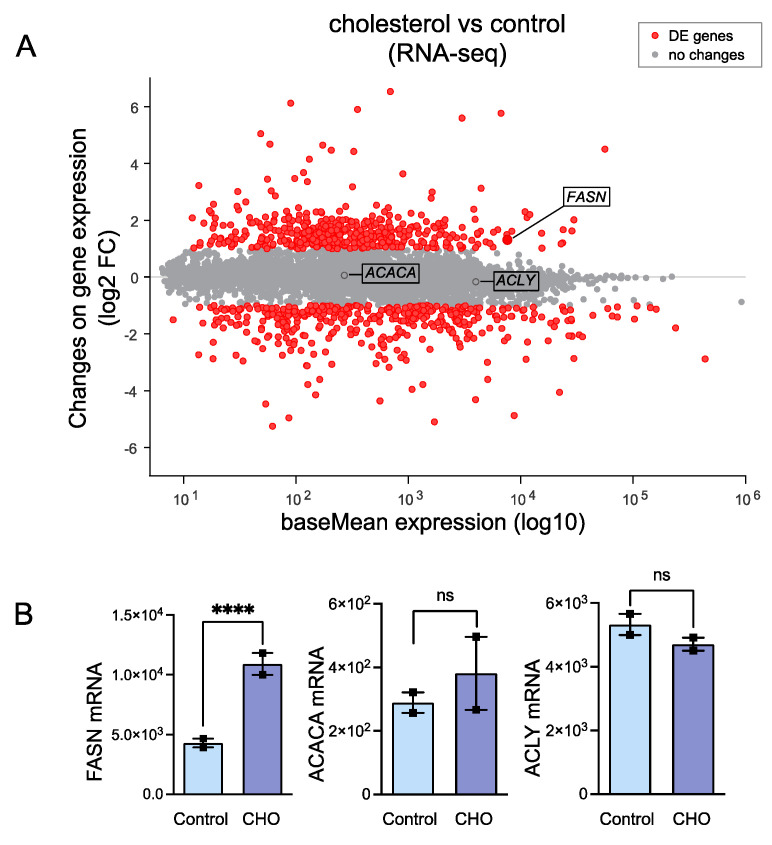
Cholesterol is shown to increase the transcriptional expression of FASN in VSMCs. Single-cell RNA-Seq counts were downloaded from the publicly available dataset GSE181362, which compares HASMCs treated with cholesterol and a vehicle for 48 h. The data were re-analyzed as described in the Section 2. (**A**) MA plot highlighting genes identified as differentially expressed (DE), marked in red. DE genes were selected based on a log2 fold change (log2 FC) of ±1, indicating a doubling or halving in expression, and an adjusted *p*-value (*p*-Adjval) of less than 0.05. (**B**) The bar graph provides specific quantification of genes from the de novo lipogenic pathway. **** *p* < 0.0001.

**Figure 3 cells-13-00658-f003:**
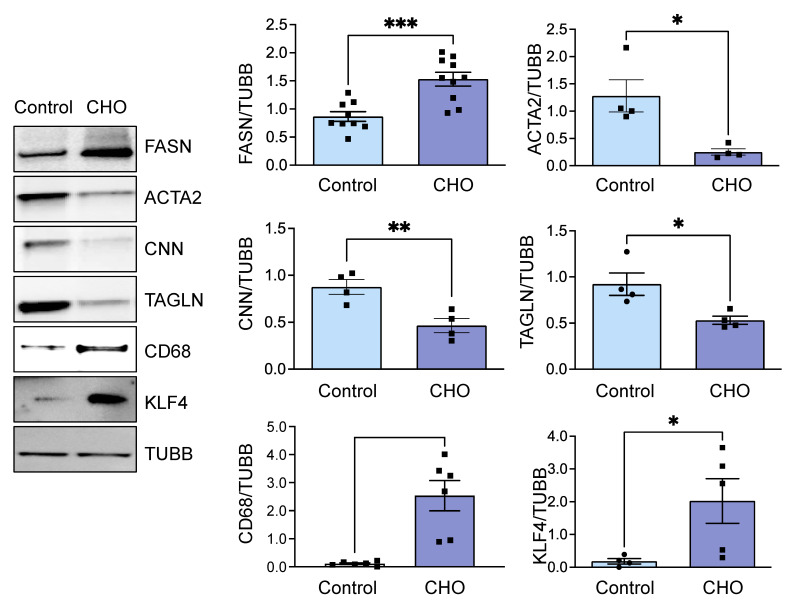
FASN upregulation accompanied a cholesterol-induced phenotypic transition in VSMC. Human aortic smooth muscle cells (SMCs) were treated with cholesterol (10 µg/mL) for 72 h. Total cell lysates were subsequently processed for western blot analysis to evaluate protein expression levels, utilizing specific antibodies. Beta-tubulin (TUBB) served as the loading control to ensure equal protein loading across samples. The images on the left display representative western blots, while the bar graphs on the right show the mean ± SEM derived from 4–9 independent experiments, illustrating the relative protein expression. * *p* < 0.05, ** *p* < 0.01, and *** *p* < 0.001.

**Figure 4 cells-13-00658-f004:**
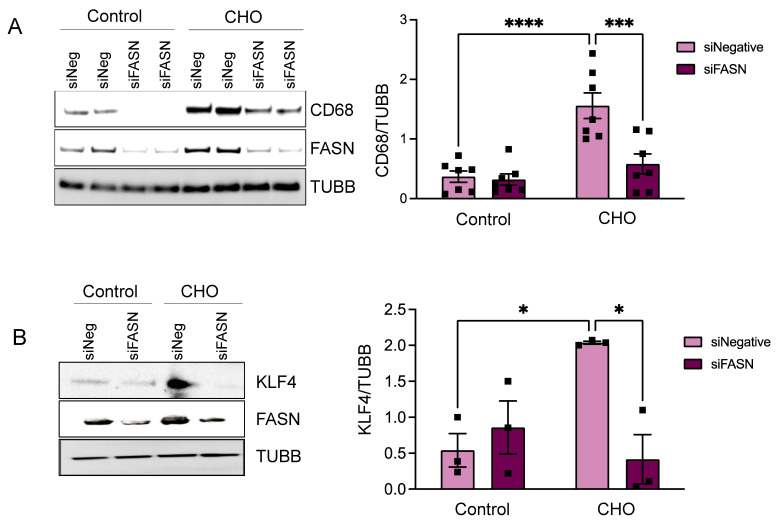
FASN deficiency decreases cholesterol-induced expression of CD68 and KLF4 in VSMCs. Human aortic SMCs were transfected with siRNA control or against FASN. After 24 h, cells were exposed to cholesterol (10 µg/mL) for 72 h. The images on the left display representative western blots, while the bar graphs on the right show the mean ± SEM derived from 3–7 independent experiments, illustrating the relative protein expression for (**A**) CD68 and (**B**) KLF4. * *p* < 0.05, *** *p* < 0.001, and **** *p* < 0.0001.

**Figure 5 cells-13-00658-f005:**
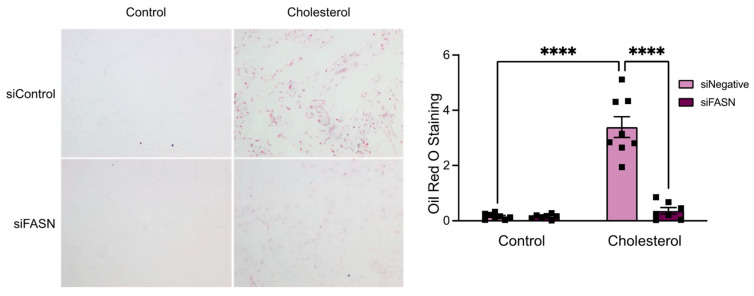
Loss of FASN inhibits cholesterol-induced intracellular lipid accumulation. Human aortic SMCs were transfected with siRNA control or against FASN. After 24 h, cells were exposed to cholesterol (10 µg/mL) for 72 h. Then, cells were fixed, and intracellular neutral lipid formation was stained using Oil Red O. Bar graphs show the mean ± SEM derived from 7 independent experiments. **** *p* < 0.0001.

**Figure 6 cells-13-00658-f006:**
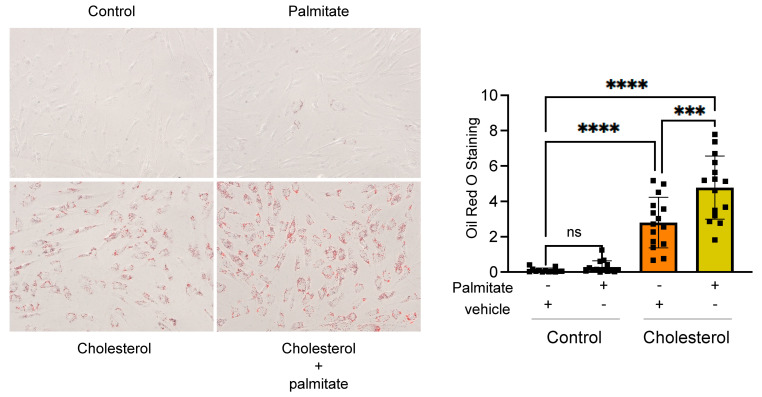
Palmitate exacerbates cholesterol-induced intracellular lipid accumulation. Human aortic SMCs were exposed to palmitate-BSA (1 µM) or BSA control in the presence or absence of cholesterol (10 µg/mL) for 72 h. Then, cells were fixed, and intracellular neutral lipid formation was stained using Oil Red O as described in the methods section. Quantitative analysis was performed on a total of 13 to 16 cells per treatment condition. The findings are consistent across two independent experiments. Bar graphs depict the mean ± SEM. *** *p* < 0.001, and **** *p* < 0.0001.

**Figure 7 cells-13-00658-f007:**
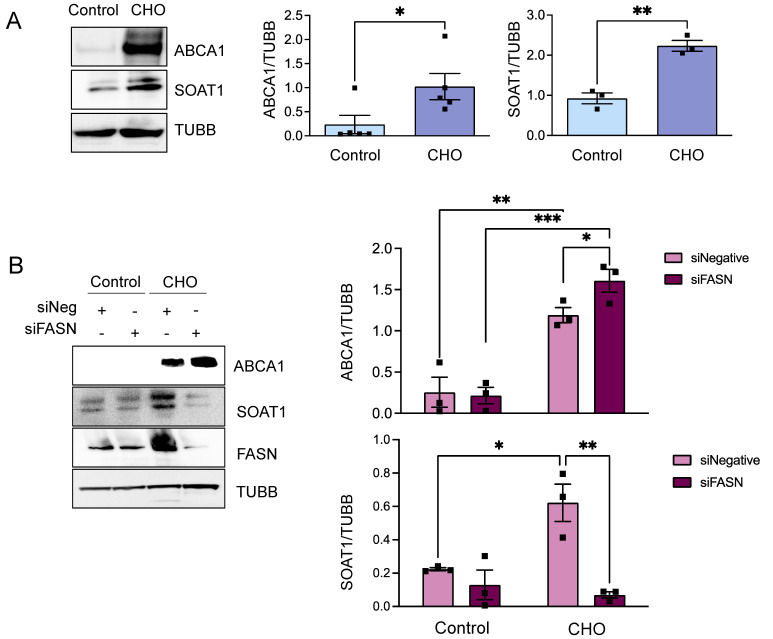
Cholesterol and FASN expression upregulate the cholesterol efflux transporter ABCA1 and the cholesterol esterase SOAT1 in VSMCs. (**A**) Human aortic SMCs were exposed to cholesterol (10 µg/mL) for 72 h; (**B**) Human aortic SMCs were transfected with siRNA control or against FASN. After 24 h, cells were exposed to cholesterol (10 µg/mL) for 72 h. Then, total cell lysates were collected for Western blot analysis, and expressions of the macrophage markers ABCA1 and SOAT1 were visualized using specific antibodies. The images on the left display representative western blots where tubulin beta was used as a loading control. Bar graphs on the right show the mean ± SEM derived from 3-4 independent experiments. * *p* < 0.05, ** *p* < 0.01, and *** *p* < 0.001.

**Figure 8 cells-13-00658-f008:**
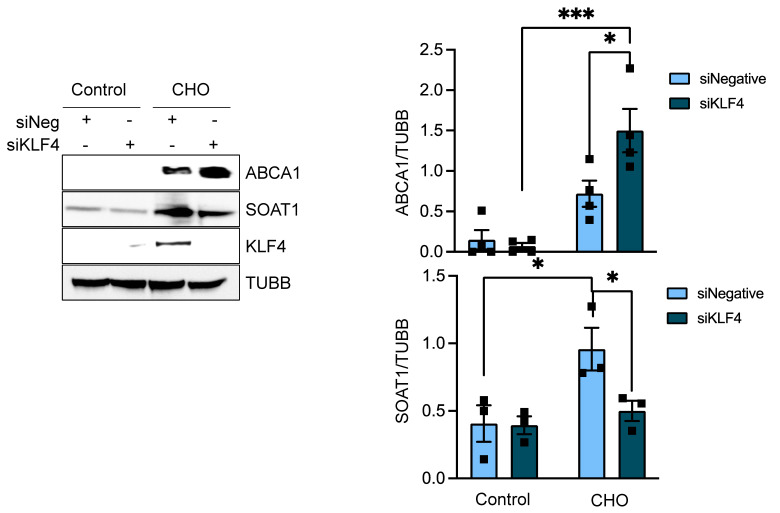
KLF4 regulates cholesterol-induced SOAT1 and ABCA1 in VSMC. Human aortic SMCs were transfected with siRNA control or against FASN. After 24 h, cells were exposed to cholesterol (10 µg/mL) for 72 h. Then, total cell lysates were collected for Western blot analysis, and protein expression was evaluated using specific antibodies. The image on the left displays representative western blots, where tubulin beta was used as a loading control. Bar graphs on the right show the mean ± SEM derived from 3-5 independent experiments. * *p* < 0.05, and *** *p* < 0.001.

**Figure 9 cells-13-00658-f009:**
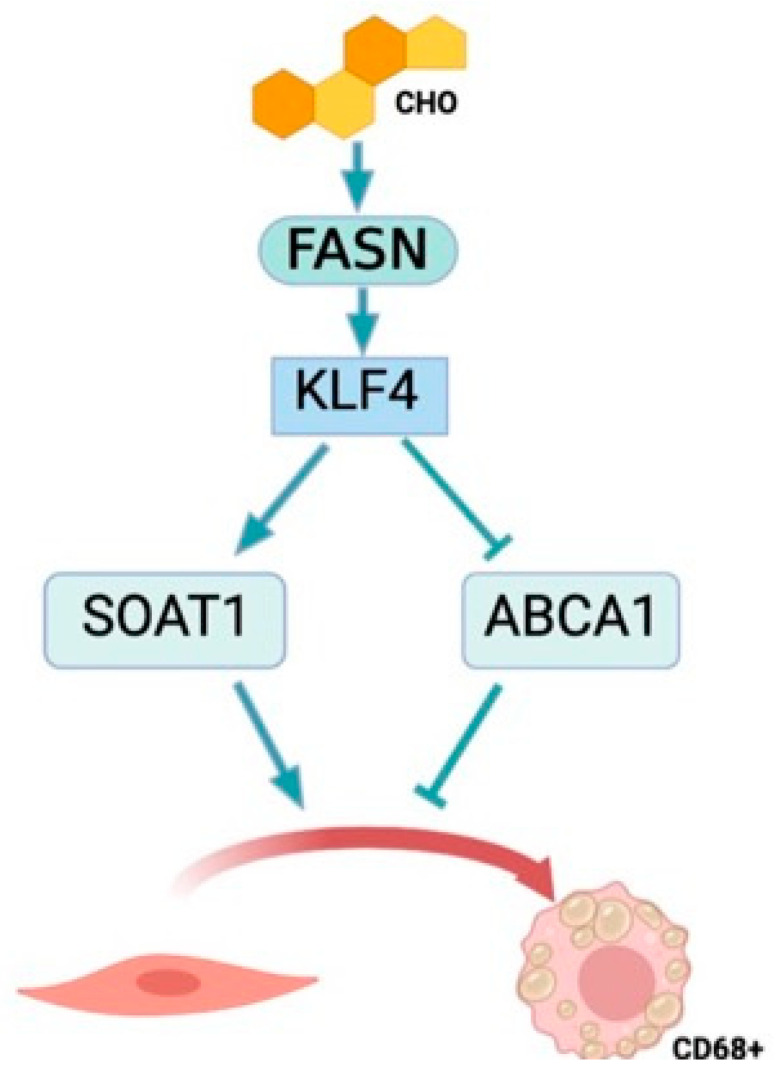
Proposed mechanism of the effects of FASN expression on foam cell formation in VSMCs. Cholesterol-mediated foam cell formation is exacerbated by FASN and KLF4 expression in VSMCs, leading to an upregulation of SOAT1 expression and a downregulation of ABCA1 expression to facilitate lipid accumulation and CD68 expression.

## Data Availability

No data to share.

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
