# Peer review of "The Role of Fatty Acid Synthase in the Vascular Smooth Muscle Cell to Foam Cell Transition"

_cells, 2024, doi:10.3390/cells13080658_

Round 1

Reviewer 1 Report

Comments and Suggestions for Authors

The study investigates the role of fatty acid synthase (FASN) in the transition of vascular smooth muscle cells (VSMCs) to foam cells, a critical process in atherosclerosis. Upon exposure to cholesterol, VSMCs undergo phenotypic changes, acquiring characteristics of foam cells. The authors found that FASN expression increases in response to cholesterol treatment in VSMCs, coinciding with the upregulation of foam cell markers. The downregulation of FASN inhibits this transition, resulting in reduced foam cell formation and lipid accumulation. Additionally, FASN deficiency affects the expression of cholesterol efflux proteins, indicating its role in cholesterol homeostasis. Moreover, the study identifies Kruppel-like factor 4 (KLF4) as a downstream effector of FASN, further elucidating the molecular mechanism underlying VSMC to foam cell transition.

Overall, the study provides a promising therapeutic target in FASN for the treatment of atherosclerosis. By elucidating the intricate molecular mechanisms underlying VSMC to foam cell transition, including the role of FASN and its downstream effector KLF4, the research offers insights into potential interventions to mitigate plaque formation and progression. Targeting FASN may not only inhibit foam cell formation but also modulate cholesterol homeostasis, suggesting a multifaceted approach to attenuate atherosclerotic burden. A few minor concerns may be addressed to enhance the quality of the manuscript.

1. Since space is not an issue, it may be better to split the immunofluorescent images in Figure 1 into separate channels for a better view of the results.

2. If it is a journal requirement, individual data points may be added in bar graphs (Figures 2B, 3, 4A, 4B, 5, 6A, 6B, and 7).

3. Some figures miss the statistical analysis results (Figure 3, 4A, 4B, 6A, 6B, and 7). Please add statistical symbols and specify the p value.

Reviewer 2 Report

Comments and Suggestions for Authors

The authors have applied an in vitro study to check the effect of FASN on VSMC differentiation to foam cell upon cholesterol treatment, the target is selected from human coronary plaque RNA-seq data, which is of interest and significance. Many concerns, however, remain to be addressed:

1.Some important methods are not clear and specific: for example, oil red O staining quantification method; RNA-seq database sample size, in vitro experiment replicates number, etc.

2. Figure 1: Suggest adding SMC and CD68 to show the cell types express FASN in the plaque.

3. If the sample size is less than 10, suggest using a bar graph with dots in all the figures.

4. Fig5, a suggestion to present a zoom-out image showing the overall lipid loading in VSMCs, is not convincing just shows one cell.

5. The data is not convincing enough to conclude that KLF4 is the downstream of FASN, as a transcription factor, KLF4 may also regulate FASN expression, is there a binding site on the FASN promoter? How is the FASN mRNA level upon KLF4 silencing?

6. Gain of function study with FASN overexpression is necessary to identify the role of FASN on VSMC deafferentation to foam cells.

Round 2

Reviewer 2 Report

Comments and Suggestions for Authors

1. The authors have revised the manuscript thoroughly.  The methods do not need to be as reagents and dilution preparation, my previous question just regarded which software was used and how the quantification was achieved, which were answered by the authors. 

2. Suggest to move the table 1 to supplemental materials.

3. The authors have missed a pertinent reference: J Pathol

. 2023 Apr;259(4):388-401. doi: 10.1002/path.6052. Epub 2023 Feb 8.

Author Response

We thank you for these suggestions. As suggested, Table 1 is now under Supplemental Materials, and we have added the very pertinent work from Cao et al. in the discussion section of the paper.
Sincerely,

Alejandra San Martin